# Hemostatic Profile of Intrauterine Growth-Restricted Neonates: Assessment with the Use of NATEM Assay in Cord Blood Samples

**DOI:** 10.3390/diagnostics14020178

**Published:** 2024-01-13

**Authors:** Eleni Karapati, Serena Valsami, Rozeta Sokou, Abraham Pouliakis, Marina Tsaousi, Alma Sulaj, Zoi Iliodromiti, Nicoletta Iacovidou, Theodora Boutsikou

**Affiliations:** 1Neonatal Department, Aretaieio Hospital, School of Medicine, National and Kapodistrian University of Athens, 11528 Athens, Greece; sokourozeta@yahoo.gr (R.S.); marina_11@windowslive.com (M.T.); alma_sulaj@hotmail.com (A.S.); ziliodromiti@yahoo.gr (Z.I.); niciac58@gmail.com (N.I.); theobtsk@gmail.com (T.B.); 2Hematology Laboratory Blood Bank, Aretaieio Hospital, School of Medicine, National and Kapodistrian University of Athens, 11528 Athens, Greece; serenavalsami@yahoo.com; 3Second Department of Pathology, University General Hospital Attikon, School of Medicine, National and Kapodistrian University of Athens, 12462 Athens, Greece; apou1967@gmail.com

**Keywords:** thromboelastometry, NATEM, ROTEM, IUGR, neonates, hemostasis, cord blood, coagulation

## Abstract

Background: Intrauterine growth restriction (IUGR) is associated with hemorrhagic and thrombotic complications during the perinatal period. Thrombocytopenia, platelet dysfunction, and prolonged standard coagulation tests are observed in this population. The aim of this study is to examine the hemostatic profile of IUGR neonates with the use of a non-activated assay (NATEM) in cord blood samples. Methods: During an 18 month period, a NATEM ROTEM assay was performed on cord blood samples of 101 IUGR neonates. A total of 189 appropriate for gestational age (AGA) neonates were used as a control group. The NATEM variables recorded include the following: clotting time (CT); clot formation time (CFT); clot amplitude at 5, 10, and 20 min (A5, A10, A20); α-angle (a°); maximum clot firmness (MCF); lysis index at 30 and 60 min (LI30, LI60); and maximum clot elasticity (MCE). Results: IUGR neonates demonstrate a hypocoagulable state, with lower A5, A10, A2, MCF, and MCE values when compared to AGA. Using multiple linear regression, we determined IUGR as an independent factor influencing all NATEM parameters (except CT and LI30) exhibiting a hypocoagulable and hypofibrinolytic profile. Platelet count was positively correlated with A5, A10, A20, MCF, alpha angle, and MCE, and negatively correlated with CFT. Conclusion: IUGR neonates appear with lower clot strength and elasticity and prolonged clot kinetics, as illustrated by ROTEM variables.

## 1. Introduction

Intrauterine growth restriction (IUGR) is characterized by the failure of the fetus to achieve its genetically determined growth potential and is a common pathology of pregnancy, affecting 10% of pregnancies worldwide [1,2]. IUGR is caused by various genetic, maternal, placental, and fetal factors [3] and is associated with stillbirth, neonatal morbidity, and mortality, especially when combined with prematurity [4]. Several short and long term complications affect growth-restricted neonates, ranging from the neonatal period to adult life [2,5,6].

The hemostatic profile of IUGR neonates is an area of wide interest, as they experience hemorrhagic, as well as thrombotic complications during the neonatal period, such as pulmonary and gastrointestinal hemorrhage, hemorrhagic neonatal stroke, and disseminated intravascular coagulopathy (DIC) [7,8,9,10,11,12,13,14]. In IUGR pregnancies, placental insufficiency is the result of thrombi formation, defective endothelium, intraplacental platelet activation, and consumption, leading to small placentas and neonatal thrombocytopenia [15,16,17]. Growth restriction and subsequent oxygen and nutrient deprivation in utero lead to a rise in erythroid progenitor cells and a subsequent downregulation of megakaryocytes, as both of them derive from the same progenitor cell, leading to hyporegenerative thrombocytopenia [18,19,20,21]. Hypo oxygenation also affects liver function due to blood flow redistribution and the brain-sparing effect [22]. Impaired liver function causes not only thrombocytopenia, as fetal hematopoiesis mainly happens in the liver [23,24,25], but also impairs the synthesis of clotting factors, resulting in the prolongation of standard coagulation tests [15,26,27,28,29].

Although platelet count and standard coagulation tests are found altered in IUGR when compared to appropriate for gestational age (AGA) infants, a correlation between laboratory findings and clinical manifestation of bleeding or thrombosis is yet to be established [13,18,29,30], leading to unnecessary platelet or fresh frozen plasma transfusions [13,18,30,31,32]. On the other hand, viscoelastic studies such as rotational thromboelastometry (ROTEM) and the thromboelastography (TEG) system portray the in vivo interaction of platelets, endothelium, clotting factors, and fibrinogen illustrating in detail every step of the formation, stabilization, and lysis of the clot [33]. The first results are available within 5–10 min for immediate therapy decisions and full qualitative results are available within 30 min. As they require a small amount of blood and provide a quick assessment of the hemostatic profile, they appear to be a useful evaluation tool for neonates. Thromboelastometry studies are scarce in the neonatal population, with reference ranges yet to be determined [34,35,36,37,38]. In regards to neonates with a specific pathology, studies on sepsis, bleeding risk, IVH, cardiac surgery, and therapeutic hypothermia after asphyxia have been conducted [39,40,41,42,43,44,45,46,47].

In non-activated assays (NATEMs), coagulation is activated without the addition of any reagent besides calcium [48]. In this respect, it reflects precisely the coagulation system in vivo [49]. It appears more sensitive to endogenous coagulation activators, such as patients with DIC or tissue factor expressed on circulating monocytes in cases of infection or sepsis [48]. Furthermore, it seems accurate in detecting any pathology regarding fibrinolysis [50,51]. In vitro studies also indicate NATEM as a useful tool in detecting endothelial damage [52]. Studies regarding NATEM are mainly focused on adults [53], assessing coagulation profile in patients with multiple myeloma [54], preeclampsia [55], hemophilia A [56,57], cirrhosis with non-neoplastic portal vein thrombosis [52], and lately COVID-19 patients [58]. As far as the neonatal population is concerned, two studies have examined NATEM in cord blood samples of healthy neonates [36,51], two studies used NATEM in order to assess coagulation statuses in critically ill neonates [59,60], but no study regarding the IUGR population was retrieved [53].

The aim of the current study is to assess the coagulation profile of IUGR neonates using a NATEM assay in cord blood samples.

## 2. Materials and Methods

This is a prospective cohort study, conducted at the Neonatal Department of Aretaieio Hospital, National and Kapodistrian University of Athens, Greece, from March 2021 to August 2022. The study was in line with all the relevant national regulations and institutional policies and was approved by the Institutional Review Board of Aretaieio Hospital, National and Kapodistrian University of Athens, Greece (Project identification code: 313/26-03-2021). An informed parental consent was obtained shortly before the enrollment of the neonate in the study.

The study population consisted of IUGR neonates born within the study’s time period. A total of 101 IUGR neonates were included: 19 preterm neonates (<37 weeks of gestation) and 82 term neonates (>37 weeks of gestation). The control group included 189 AGA neonates, who were formerly studied and published by our research team [36]. Inclusion criteria for the IUGR group was an estimated fetal weight below the 10th centile and a definite cause for fetal growth restriction, affecting fetal biometric measurements and Doppler flow findings [61]. Birth weight centile was calculated using the gestation-related optimal weight (GROW) program [62,63,64]. This computer program customizes birth weight centile taking into account factors that determine birth weight such as gestational age, gender, booking maternal weight, maternal height, ethnicity, and parity. Exclusion criteria included chromosomal abnormalities, genetic disorders, metabolic syndromes of the fetus, any congenital or inherited coagulation disorder such as hemophilia, and congenital or perinatal infections such as early onset sepsis or chorioamnionitis.

Shortly after clamping the umbilical cord, we collected cord blood samples with the use of 21G needle. They were immediately transferred into 3.5 mL 9NC coagulation sodium citrate 3.2% containing VACUETTE^®^ TUBE, Greiner Bio-One GmbH, Kremsmünster Austria. Blood samples were carefully assessed and discarded if fibrin clots were detected. Afterwards, the sample was inverted five times in order for any sediment to be resuspended and incubated for 2–5 min at 37 °C. Whole blood (300 μL) was analyzed on the ROTEM^®^ delta analyzer (Tem Innovations GmbH, Munich, Germany) using the NATEM assay immediately after sample collection. The ROTEM test was executed using the automated pipette programs as per manufacturer’s guidelines. Clot formation was induced by adding 20 μL of 0.2 M calcium chloride solution (star-TEM^®^ 20 reagent, Tem Innovations GmbH, Munich, Germany). After adding the reagent to the cup, it was adequately mixed with 300 μL of whole blood anticoagulated with 0.109 mol/L trisodium citrate (9:1, *v*/*v* blood anticoagulant, Greiner Bio-One GmbH, Kremsmünster, Austria). The assay ran for at least 60 min after clot lysis at 30 min. NATEM variables recorded included the following: clotting time (CT, seconds), the time passed from the beginning of measurement until the formation of a clot 2 mm in amplitude; clot formation time (CFT, seconds), the time between 2 mm and 20 mm of clot amplitude; clot amplitude at 5, 10, and 20 min after CT (A5, A10, A20); α-angle (a°), the angle between the central line (*x*-axis) and the tangent of the clotting curve at the amplitude point of 2 mm, reflecting the clot kinetics and measured in degrees; maximum clot firmness (MCF, mm),the widest amplitude of main body of trace; lysis index at 30 and 60 min (LI30, LI60, %), the percentage of remaining clot stability proportionate to the MCF following the 60-min observation period after CT indicating the speed of fibrinolysis; and maximum clot elasticity(MCE = 100 × MCF/(100—MCF)).

All neonates received 1 mg of vitamin K intramuscularly (IM) and were followed up closely until discharge and any complication was noted. Data concerning gestational age, birth weight, gender, mode of delivery, birth centile, Doppler pulsatility index (PI), Apgar score, pH of the umbilical cord and hemoglobin, and platelet count of the umbilical cord were recorded. Data in regard to maternal coagulopathy, use of anticoagulant medication, smoking during pregnancy, maternal BMI, and the presence of preeclampsia or hypertension were reported. Data about neonatal intensive care unit (NICU) stay and neonatal complications (such as sepsis, hypoglycemia, neonatal respiratory distress syndrome (RDS), thrombosis, or hemorrhagic events) were documented as well.

All IUGR cases included were monitored closely with prenatal scans and Doppler studies of the PI of the uterine, umbilical, and middle cerebral artery during pregnancy. In regard to uterine and umbilical arteries, the mean PI values were progressively increased and detected at the upper limits for gestational age (between the 90th and 95th percentile) in two-thirds of the cases. In the remaining one-third, the PI values where increased and found above the 95th percentile for gestational age. With respect to middle cerebral arteries, the PI values were in the lower limits for gestational age, demonstrating the brain-sparing effect occurring in IUGR fetuses [65].

The placentas were small (less than 400 g) and infarcted. All IUGR cases were asymmetrical. In 19 cases of IUGR fetuses, mothers smoked more than 10 cigarettes per day. Nine IUGR neonates were born from twin pregnancies. In 17 cases, the mother suffered from hypertension of pregnancy, with 9 of them developing severe preeclampsia during follow up. In regards to diabetes, 3 mothers had type 2 diabetes and 14 mothers were diagnosed with gestational diabetes during pregnancy. Fifteen mothers of IUGR infants were obese, with BMIs over 30. In five cases, the mother was diagnosed with thrombophilia, resulting in small and infracted placenta. In the remaining 19 cases, oligohydramnios and small infracted placenta was present. Furthermore, the cephalization index (CI) (ratio of head circumference to body weight) [66] and the HC/AC ratio (ratio of head circumference to abdominal circumference) [67] were additionally utilized as measures of growth restriction in utero.

### Statistical Analysis

The statistical analysis was performed using the SAS for Windows version 9.4 software platform (SAS Institute Inc., Cary, NC, USA) (DiMaggio, 2013; SAS Institute, 2014). Descriptive values were expressed as median and Quartile 1 (Q1) to Quartile 3 (Q3) range. Comparisons between the groups for the qualitative parameters were made using the chi-square test (and if required a Fisher exact test was performed). For the continuous parameters normality was not possible to be ensured (as evaluated by the Shapiro–Wilk method), therefore not parametric tests were applied, in particular the Mann Whitney U test and the Kruskal–Wallis test (if more than two groups). Focusing on the IUGR neonates, we used the Spearman correlation coefficient test to correlate NATEM parameters with laboratory and demographic data in this group. Finally, in order to investigate whether IUGR is an independent factor causing hemostatic alterations, as portrayed through NATEM parameters, an adjustment regarding other confounding factors was conducted, using multivariate analysis (multiple linear regression). Specifically, we used each calculated NATEM parameter as dependent variable and the group (i.e., IUGR or AGA), gestational age, gender, and delivery mode as independent variables. The chosen covariates represent the most common confounding factors that could affect the NATEM results. The significance level (*p*-value) was set to 0.05 and all tests were two-sided.

## 3. Results

During the study period, 101 IUGR and 189 control AGA neonates were recruited. Demographic data of our study population are depicted in Table 1. Delivery method differs significantly, with caesarian section being the method of choice regarding IUGR neonates (80.2% vs. 61.38%, *p*: 0.001). The gestational age and birth weight also differ significantly; median gestational age of the IUGR neonate is 38 weeks vs. 39 weeks for AGA (38 vs. 39 weeks of gestation, *p*: <0.0001). Median birth weight of the IUGR group is 2540 vs. 3330 for the AGA group (*p*: <0.0001). As for the mothers of IUGR newborns, they appear to have higher BMI values (24 vs. 22, *p*: 0.005) and statistically significant higher odds of gestational or type II diabetes (16% vs. 0%, *p*: <0.0001), hypertension (17% vs. 0%, *p*: <0.0001), and preeclampsia (9% vs. 0%, *p*: <0.0001). With respect to the perinatal period, IUGR neonates have statistically significant lower values of umbilical cord pH (7.3 vs. 7.4, *p*: <0.0001) and higher rates of NICU admission (6% vs. 0%, *p*: 0.0016), RDS (4% vs. 0%, *p*: 0.014), hypoglycemia (38% vs. 7%, *p*: <0.0001), and central nervous system (CNS) bleeding (8% vs. 0%, *p*: 0.0002). No incidence of thrombosis was noted in both groups.

Comparison of NATEM parameters between the AGA and IUGR group is portrayed in Table 2 and graphically illustrated in Figure 1. IUGR neonates have lower A5 (*p*: 0.013), A10 (*p*: 0.004), A20 (*p*: 0.004), MCF (*p*: 0.008), and MCE (*p*: 0.01) compared to AGA. No significant statistical differences were noted in CT, CFT, alpha angle, LI30, and LI60 between the AGA and the IUGR group.

Following multivariate analysis IUGR was established as an independent factor affecting all NATEM parameters (except CT and LI30), as summarized in Table 3, by lower values of A5, A10, A20, MCF, alpha angle, and MCE in IUGR compared to AGA neonates and higher values of CFT and LI60. Regarding the other factors, gestational age seems to affect fibrinolysis and clot elasticity with higher LI30, LI60, and lower MCE. Gender and delivery mode did not have any effect in NATEM parameters.

Within the IUGR group consisting of 101 neonates, we evaluated the association of gestational age, birth weight, pH, platelet count, and hemoglobin of the umbilical cord with NATEM parameters using Spearman correlation coefficients, as presented in Table 4. pH is negatively correlated with LI30 and LI60 and gestational age is positively correlated with LI30 and LI60. Birth weight seems to be weakly positively correlated with LI60. Platelet count is positively correlated with A5, A10, A20, MCF, alpha angle, and MCE and negatively correlated with CFT. Hemoglobin is negatively correlated with A5, A10, A20, MCF, alpha angle, and MCE and positively correlated with CFT.

## 4. Discussion

To the best of our knowledge, this is the first study examining the coagulation profiles of IUGR neonates using ROTEM variables of the NATEM assay. When compared to controls, IUGR neonates have lower A5, A10, A20, MCF, and MCE compared to AGA, designating a hypocoagulable profile. Using multiple linear regression, we determined IUGR as an independent factor causing hemostatic alterations.

Growth-restricted infants are characterized by lack of in utero oxygenation and nutrients leading to low birth weight [1,3,68]. Although IUGR per se is not an indication for caesarian section, when combined with a non-reassuring fetal status such as early abnormal Doppler findings, oligohydramnios, and extreme prematurity, primary or urgent caesarian section is the delivery mode of choice [68]. As indicated by our data, our IUGR study group has lower pH values in the umbilical cord, reflecting perinatal stress and a non-reassuring fetal status, leading to a high percentage of requiring a caesarian section. Diabetes (type II or gestational), hypertension, preeclampsia, and maternal obesity are leading maternal factors causing growth restrictions [1,3], which is in line with our population. Fetal compromises and the possibility of stillbirth leads to earlier deliveries and higher odds of prematurity [1,4,68]. Growth restrictions especially when combined with prematurity result in higher chances of NICU admission, with common postnatal complications being hypoglycemia and RDS, which is in agreement with our findings [2,3,4,7].

The hemostatic system of the newborn is an age-developing system and differs significantly from that of the adult not only in quality, but also in quantity of clotting and the fibrinolytic factors involved [69,70,71]. Nevertheless hemostasis in a healthy newborn is functionally balanced, with neither thrombotic nor hemorrhagic predisposition, an equilibrium that is disrupted in pathological conditions [72]. As depicted in our study, IUGR infants appeared with lower A5, A10, A20, MCF, and MCE values when compared to the AGA control group. MCF, as illustrated by the maximum amplitude in the NATEM assay, reflects maximal clot firmness attained during testing and is mainly dependent on platelet count and function, fibrin concentration and polymerization, factor XIII activity, and colloids [48]. A5, A10, and A20, portraying the amplitude of clot firmness in 5, 10, and 20 min, respectively, are correlated with MCF. They are used in everyday practice when fast decision-making must be accomplished regarding the hemostatic status of the patient. MCE reflects maximal clot elasticity and although associated with MCF, it expresses the platelet component of clot firmness [48].

Conclusively, IUGR infants present with a hypocoagulable profile, as depicted by diminished clot firmness and elasticity. In our study, multiple linear regression showed that IUGR is an independent factor affecting NATEM parameters. IUGR is a known cause of early onset thrombocytopenia [18,73,74,75,76,77]. It usually resolves spontaneously within the first two weeks of life, without requiring specific treatment, as it is usually mild [13,30,31]. The degree and duration of thrombocytopenia is correlated with birth weight centile [13,18,30,78] and Doppler flow abnormalities, such as umbilical arterial pulsatility index (UA-PI) and absence of end-diastolic velocity (AEDV) [79,80,81,82,83]. Besides platelet count, platelet dysfunction is also reported in IUGR neonates. Kollia et al. [84] confirmed platelet hyporesponsiveness in IUGR neonates using PFA-100, which is in line with our observations. A positive correlation of platelet count with A5, A10, A20, MCF, alpha angle, and MCE was observed in our study. As platelets are key determinants of the affected NATEM values [85], our results could be possibly attributed to platelet dysfunction, since thrombocytopenia was not affirmed. Furthermore, as aforementioned, MCF is also dependent on fibrin concentration and polymerization. Data regarding fibrinogen levels in IUGR infants are contradicting and scarce. Several studies reported lower fibrinogen in IUGR neonates [27,86,87,88], although others fail to confirm this observation [15,89]. Lower fibrinogen values could also have an impact on lower values of maximal clot amplitude, as revealed by our results.

After adjusting for confounding factors, we observed that IUGR was additionally associated with lower alpha angle and higher CFT and LI60 values. CFT and alpha angle reflect clot kinetics; CFT the time needed for creating a firm and stable clot between 2 and 20 mm and alpha angle the speed of clot formation, portraying the united contribution of fibrinogen and platelets to clot strength [48]. As both of those values are dependent on platelet count and function, fibrinogen concentration and polymerization, clotting factors, and thrombin generation, lower values found in the IUGR group underlines the hypocoagulable profile of that population, as aforementioned. Prolonged PT, INR, and APTT have been observed in IUGR infants, mainly attributed to reduced production of clotting factors [15,26,27,28,29,87]. As clotting factors do not cross the placenta, they are produced mainly in the fetal liver, a process affected in IUGR fetuses due to in utero hypo oxygenation and subsequent fetal liver dysfunction [90]. LI60 represents the clot remaining at 60 min, meaning that higher LI60 values spotted in the IUGR neonates of our study designate a hypo fibrinolytic profile, compared to the control group. Studies regarding the fibrinolytic system of IUGR neonates are rare and contradicting. Fuse et al. [91] reported lower levels of plasminogen and alpha2-plasmin inhibitor in small for gestational age (SGA) neonates vs. AGA and a hypofibrinolytic profile in the mothers of SGA infants. In contrast with our findings, Ekelund et al. [89] remarked no difference in the fibrinolytic activity of SGA vs. AGA infants, although he reported lower plasminogen levels in ill SGA term infants. Finally, Mitsiakos et al. [29,87] reported higher levels of tissue plasminogen activator (t-PA) in SGA full term infants and higher levels of both t-PA and plasminogen activator inhibitor-1 in preterm SGA neonates.

Focusing on the IUGR group, we detected a positive correlation between gestational age and LI30 and LI60 using Spearman correlation coefficients. Similar data were reported by Theodoraki et al. [37] and Sulaj et al. [36] when studying full-term neonates with the use of the INTEM/EXTEM and NATEM assay, respectively. This observation is explained by the lower level of fibrinolytic inhibitor proteins during prematurity [92]. IUGR neonates are susceptible to perinatal hypoxia/asphyxia, hypothermia, meconium aspiration, and hypoxic-ischaemic encephalopathy [93]. Umbilical cord artery pH indicates the degree of metabolic acidosis and as such, birth hypoxia. It is acknowledged as a mandatory criterion for the definition of birth asphyxia [94,95]. Thromboelastometry studies focusing on neonates with perinatal hypoxia values are rare [96]. Konstantinidi et al. detected higher LI60 values in hypoxic and asphyxiated infants when compared to controls [97]. In experimental studies on hypoxic mice, an upregulation regarding plasminogen activator inhibitor-1 (PAI-1) and downregulation of plasminogen activators was noted, leading to fibrinolytic suppression [98]. In our study population, a negative correlation of the umbilical cord pH with LI30 and LI60 was revealed, while unexpectedly birth weight within the IUGR group was only found weakly positively correlated with LI60. One possible explanation for this phenomenon is that hematological alterations noted using ROTEM are caused by the state of growth restriction per se and not the final birth weight. Birth weight only provides a vague estimation of the degree of growth restriction, as it greatly depends on genetic potential of the fetus. The time of insult, Doppler abnormalities found at umbilical, uterine, and middle cerebral artery, and a combination of fetal anthropometric measures (weight, head, and abdominal circumference) are key determinants of the degree of growth restriction in utero [68]. Additionally, birth weight is widely dependent on gestational age. Since gestational age is positively correlated with LI60, prematurity per se could account for the aforementioned correlation. In utero oxygen deprivation in growth-restricted fetuses leads to a compensatory rise in red cell production, high hematocrit values, and nucleated red blood cells. As platelets and red blood cells originate from the same multipotent stem cell, this shift in favor of erythroid cell line causes thrombocytopenia. As outlined above, platelet count is found positively correlated with A5, A10, A20, MCF, alpha angle, and MCE and negatively correlated with CFT. Hemoglobin was inversely correlated with the same NATEM parameters, relating a hypocoagulable profile in IUGR neonates with higher hemoglobin values. Red blood cells appear to affect the hemostatic system by interacting with platelets, endothelium, fibrinogen, and blood viscosity [99]. Our observation is in line with the studies of Theodoraki et al. [37] and Westbury et al. [100]. One possible explanation of this phenomenon is that high hematocrit downregulates fibrinogen activity by diminishing fibrinogen concentration in whole blood [100].

Only one ROTEM study assessing the hemostatic profile of IUGR neonates was retrieved, detecting no differences between IUGR and AGA neonates in contrast with our findings [101]. This divergence could be attributed to the different ROTEM assay used (EXTEM vs. NATEM). As the NATEM assay is used without any additional reagent, it reflects more accurately the hemostatic process in vivo [49,50]. Moreover the sample used is dissimilar; our study used cord blood, whereas the study by Sokou et al. used neonatal arterial blood drawn from the 2nd to 7th day of life. Additionally, our study focused exclusively on IUGR neonates while the other study’s population consisted of SGA infants, with 86,7% of them being classified as IUGR. Although SGA infants have a birth centile below the 10th, only a cluster of them present with intrauterine growth restrictions and abnormal Doppler findings [3].

The hypocoagulable predisposition of IUGR infants, as depicted by prolonged PT, INR, and APTT, is noted in studies regarding standard coagulation tests [15,26,27,28,29,88]. Nevertheless, the link between laboratory assessment and clinical manifestation is yet to be determined [29,87,90]. IUGR is associated with hemorrhagic complications such as pulmonary hemorrhage, gastrointestinal bleeding, and hemorrhagic neonatal stroke [7,10,11,13]. Data regarding the occurrence of IVH are controversial. Rocha et al. [102], Liu et al. [7], and Duppré et al. [103] recognized IUGR as a risk factor for IVH, yet Fustolo-Gunnink et al. [30] found no link between growth restriction and IVH. Using cranial ultrasound, we were able to detect eight mild cases of intracranial bleeding in the IUGR study population (intraventricular hemorrhage grade I), a higher percentage when compared to control. Because of the small number of bleeding incidences in our study, an analysis to correlate them with the hemostatic derangement—as depicted by NATEM parameters in the IUGR population—could not be performed. Thus, it is not clear whether those incidences are attributed to the hypocoagulable profile of IUGR neonates as noted in our study, or to the higher odds of prematurity observed in this population.

The study’s limitations should be taken into account. Due to delayed cord clamping or cord milking practices in our institution a small amount of blood was used for ROTEM analysis and complete blood count. As a result, standard coagulation tests could not be performed and thus correlated to NATEM parameters. Moreover, it should be taken into account that our study’s sample of choice was umbilical cord blood due to minimal handling practices. Laboratory results from umbilical cord and neonatal blood seem comparable and umbilical cord blood is used more frequently in the NICU for complete blood count and sepsis evaluation [104,105]. In this respect, larger cohort studies are required in order to confirm our results.

## 5. Conclusions

To the best of our knowledge, this is the first study evaluating the hemostatic profile of IUGR neonates using a NATEM assay in cord blood samples. IUGR neonates present a hypocoagulable and hypofibrinolytic profile, with lower clot strength and elasticity and prolonged clot kinetics. Although larger studies are needed to verify our data, our study underlines the coagulation derangement in this sensitive population, aiming at early detection of these disorders and optimal handling.

## Figures and Tables

**Figure 1 diagnostics-14-00178-f001:**
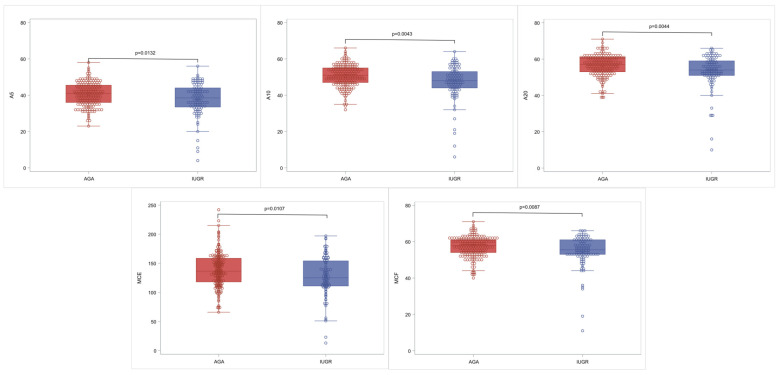
Characteristic box and whisker plots for the statistically significant NATEM parameters. In each diagram, the box limits indicate the lower (1st) and higher (3rd) quartiles (Q1 and Q3, respectively), the horizontal lines within the boxes indicate median value, while the limits of the whiskers indicate minimum and maximum values after excluding outliers. The circles correspond to actual measurements and outliers are outside the whisker limits.

**Table 1 diagnostics-14-00178-t001:** Demographic, clinical, and laboratory data of the study population.

Characteristic	AGA (*N* = 189) Median (Q1–Q3) or *N* (%)	IUGR (*N* = 101) Median (Q1–Q3) or *N* (%)	*p*
Delivery method	CS (116/61.38%) VD (73/38.62%)	CS (81/80.2%) VD (20/19.8%)	0.0010
Gender	Female (86/45.5%) Male (103/54.5%)	Female (51/50.5%) Male (50/49.5%)	0.4596
Gestational age (weeks)	39 (39–40)	38 (37–39)	<0.0001
Birth weight (gram)	3330 (3140–3530)	2540 (2280–2750)	<0.0001
Smoking	No (161/85.18%) Yes (28/14.82%)	No (80/79.21%) Yes (21/20.79%)	0.2492
Maternal BMI	22 (20.5–25)	24 (22–27)	0.0057
Thrombophilia	No (178/94.18%) Yes (11/5.82%)	No (95/94.06%) Yes (6/5.94%)	1.0000
Diabetes	No (189/100%) Gestational (0/0%) Type II (0/0%)	No (77/82.8%) Gestational (13/13.98%) Type II (3/3.23%)	<0.0001
Hypertension	No (189/100%) Yes (0/0%)	No (84/83.17%) Yes (17/16.83%)	<0.0001
Preeclampsia	No (189/100%) Yes (0/0%)	No (92/91.09%) Yes (9/8.91%)	<0.0001
pH (Umbilical Cord)	7.4 (7.3–7.4)	7.3 (7.3–7.4)	<0.0001
NICU admission	No (189/100%) Yes (0/0%)	No (95/94.06%) Yes (6/5.94%)	0.0016
Sepsis	No (189/100%) Yes (0/0%)	No (100/99.01%) Yes (1/0.99%)	0.3483
Respiratory distress syndrome (RDS)	No (189/100%) Yes (0/0%)	No (97/96.04%) Yes (4/3.96%)	0.0141
Hypoglycemia	No (182/96.3%) Yes (7/3.7%)	No (63/62.38%) Yes (38/37.62%)	<0.0001
CNS bleeding	No (189/100%) IVH (0/0%)	No (93/92.08%) IVH (8/ 7.92%)	0.0002
Thrombosis	None	None	NA
Platelet count (umbilical cord)	No data	276,400 (231,900–316,100)	NA
Hemoglobin (umbilical cord)	No data	15.6 (13.8–17.2)	NA

Abbreviations: IUGR—intrauterine growth restriction; AGA—appropriate for gestational age; CS—cesarean section; VD—vaginal delivery; BMI—body mass index; NICU—neonatal intensive care unit; CNS—central nervous system;; IVH—intraventricular hemorrhage grade I; NA—not applicable.

**Table 2 diagnostics-14-00178-t002:** Comparison of NATEM parameters between AGA and IUGR neonates.

	AGA (*N* = 189)	IUGR (*N* = 101)	
NATEM Parameter	Median (Q1–Q3)	Median (Q1–Q3)	*p*
CT	322 (250–391)	309.5 (259–414.5)	0.8423
A5	41 (36–45.5)	38.5 (33.5–44)	0.0132
A10	51 (47–55)	48 (44–53)	0.0043
A20	57 (53–61)	54 (51–59)	0.0044
CFT	97 (80–127)	106.5 (81–136)	0.1871
MCF	58 (54–61)	55.5 (53–61)	0.0087
alpha angle	71 (65–74)	69 (64–74)	0.1807
LI30	100 (99–100)	100 (99–100)	0.8171
LI60	93 (91–95)	93 (91–96)	0.0613
MCE	136.5 (118–158.5)	125 (111–154)	0.0107

Abbreviations: CT—clotting time (seconds); CFT—clot formation time (seconds); A5, A10, A20, clot amplitude at 5, 10, and 20 min, (mm); MCF—maximal clot firmness (mm); LI30, LI60, and lysis index at 30 and 60 min (%); α-angle—alpha angle (α); MCE—maximum clot elasticity; NATEM—non-activated rotational thromboelastometry; AGA—appropriate for gestational age; IUGR—intrauterine growth restriction; *N*—number of cases; Q1—first quartile; Q3—third quartile.

**Table 3 diagnostics-14-00178-t003:** Results of multiple linear regression analysis for NATEM parameters as dependent variables and IUGR, gestational age, gender, and delivery mode as independent variables.

Parameter	CT			A5			A10			A20			CFT		
	Beta	SE	*p*	Beta	SE	*p*	Beta	SE	*p*	Beta	SE	*p*	Beta	SE	*p*
IUGR (ref = AGA)	−4.71	12.56	0.7077	−2.79	0.99	0.005	−3.15	0.99	0.0016	−2.80	0.93	0.0029	15.83	7.14	0.0274
Gestational age	−0.34	4.19	0.9361	0.11	0.33	0.7491	0.15	0.33	0.6599	0.33	0.31	0.2873	−0.31	2.36	0.8967
Gender Female (ref = male)	−16.22	11.06	0.1435	0.92	0.87	0.2903	1.00	0.87	0.2516	1.08	0.82	0.1896	−9.52	6.23	0.1279
Delivery mode CS (ref = VD)	10.30	12.31	0.4037	−0.23	0.97	0.8096	0.05	0.97	0.956	0.44	0.91	0.6326	2.20	6.97	0.7521
	**MCF**			**a** **lpha angle**			**LI30**			**LI60**			**MCE**		
**Parameter**	**Beta**	**SE**	** *p* **	**Beta**	**SE**	** *p* **	**Beta**	**SE**	** *p* **	**Beta**	**SE**	** *p* **	**Beta**	**SE**	** *p* **
IUGR (ref = AGA)	−2.39	0.89	0.0075	−2.28	0.94	0.0155	0.07	0.12	0.538	1.50	0.42	0.0004	−8.38	4.15	0.0444
Gestational age	0.40	0.30	0.1807	−0.19	0.31	0.547	0.14	0.04	0.0004	0.53	0.14	0.0002	2.81	1.38	0.0432
Gender Female (ref = male)	0.91	0.78	0.246	1.35	0.82	0.1019	0.03	0.10	0.7904	−0.14	0.37	0.7008	4.85	3.66	0.1855
Delivery mode CS (ref = VD)	0.38	0.87	0.6648	−0.34	0.92	0.7079	0.11	0.12	0.3256	−0.06	0.41	0.8757	−0.33	4.07	0.9361

Abbreviations: CT—clotting time (seconds); CFT—clot formation time (seconds); A5, A10, A20, clot amplitude at 5, 10, and 20 min, (mm); MCF—maximal clot firmness (mm); LI30, LI60, and lysis index at 30 and 60 min (%); α-angle—alpha angle (α); MCE—maximum clot elasticity; NATEM—non-activated rotational thromboelastometry; IUGR—intrauterine growth restriction; CS—cesarean section; VD—vaginal delivery; AGA—appropriate for gestational age.

**Table 4 diagnostics-14-00178-t004:** Spearman correlation coefficients of NATEM parameters with laboratory and demographic data in the IUGR group.

	CT	A5	A10	A20	CFT	MCF	Alpha Angle	LI30	LI60	MCE
Gestational age (weeks)	−0.02 0.8575 100	0.05 0.6222 100	0.07 0.4807 100	0.11 0.2653 100	−0.01 0.9015 98	0.13 0.2104 100	−0.02 0.8617 99	0.28 0.0048 100	0.30 0.0024 100	0.14 0.1706 100
Birth weight (gram)	−0.03 0.768 100	0.13 0.2088 100	0.14 0.1751 100	0.16 0.122 100	−0.07 0.467 98	0.15 0.1475 100	0.05 0.6569 99	0.17 0.0904 100	0.21 0.0373 100	0.15 0.1249 100
pH (Umbilical Cord)	−0.17 0.0951 95	0.10 0.3328 95	0.09 0.3815 95	0.05 0.6632 95	−0.14 0.189 93	−0.02 0.8385 95	0.10 0.36 94	−0.31 0.0026 95	−0.34 0.0007 95	−0.02 0.8487 95
Platelet count (Umbilical Cord)	−0.22 0.2855 25	0.50 0.011 25	0.50 0.0116 25	0.58 0.0024 25	−0.45 0.0237 25	0.60 0.0014 25	0.51 0.01 25	−0.21 0.3245 25	−0.17 0.4126 25	0.62 0.001 25
Hemoglobin (Umbilical Cord)	0.19 0.3589 25	−0.75 <0.0001 25	−0.77 <0.0001 25	−0.63 0.0008 25	0.63 0.0007 25	−0.57 0.0029 25	−0.58 0.0022 25	0.38 0.0645 25	0.18 0.3775 25	−0.58 0.0025 25

Abbreviations: CT—clotting time (seconds); CFT—clot formation time (seconds); A5, A10, A20, clot amplitude at 5, 10, and 20 min, (mm); MCF—maximal clot firmness (mm); LI30, LI60, and lysis index at 30 and 60 min (%); α-angle—alpha angle (α); MCE—maximum clot elasticity; NATEM—non-activated rotational thromboelastometry; IUGR—intrauterine growth restriction.

## Data Availability

Date are available from the corresponding author upon a reasonable request.

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
