# Peer review of "Hemostatic Profile of Intrauterine Growth-Restricted Neonates: Assessment with the Use of NATEM Assay in Cord Blood Samples"

_diagnostics, 2024, doi:10.3390/diagnostics14020178_

Round 1

Reviewer 1 Report

Comments and Suggestions for Authors

It is a thorough and extensive investigation of the coagulation system in a woman with a low blood pressure and intrauterine growth retardation (IUGR). 101 children with IUGR were included, along with a control group of 189 children. Special methods were used to measure all possible parameters in IUGR children and the control group of children. The differences were then tested using appropriate statistical methods and presented as a p value. In my opinion, the entire research was done thoroughly with respect for all scientific categories. The results of the research are an incentive for other doctors with additional results because this is, apparently, the first such extensive research. The paper deserves publication

Author Response

Dear Reviewer,

Thank you very much for taking the time to review this manuscript. We are honoured by your kind comments and thankful for considering our paper for publication. 

Reviewer 2 Report

Comments and Suggestions for Authors

The study explored the hemostatic profile at non-activated assay (NATEM) in 101 IUGR neonates (19 preterm and 82 term neonates) and compared the results to those obtained in 189 infants appropriate for gestational age (AGA). Cord blood samples were utilized to perform the assay. The authors concluded that IUGR neonates exhibit a hypo-coagulative and hypo-fibrinolytic state, as demonstrated by abnormal values of parameters connected to the clot amplitude, maximal clot firmness, clot elasticity, and clot lysis. These results were mostly ascribed to the abnormal platelet function reported in these patients.

The topic is original and these observations add some novelty to the specific field. The overall impact of the reported findings in the clinical management of IUGR  patients remains to be established.

I have some comments.

1.       1.       Materials and Methods could be better reorganized: the study design should be better detailed, specifying inclusion and exclusion criteria, procedures, outcomes, and collected data.  

2.       The authors report a higher incidence of intracranial bleeding in the IUGR study population than in controls. There was also a higher incidence of platelet or plasma transfusions among IUGR neonates? 

3.       The manuscript is clear enough, but sometimes it appears redundant. For example, data in Figure 1 are also presented in Table 1.

4.       It is unclear if IUGR and AGA groups were both included in the multivariate regression analysis. How did the authors select the covariates?  

Author Response

Response to Reviewer 2 Comments

Thank you very much for taking the time to review our manuscript and for your kind notes and comments. Please find the detailed responses below and the corresponding corrections using track changes in the re-submitted manuscript.

  1. Materials and Methods could be better reorganized: the study design should be better detailed, specifying inclusion and exclusion criteria, procedures, outcomes, and collected data.  

Response 1: Thank you for pointing this out. We have modified the “Materials and Methods” section in our manuscript accordingly. Revised section can be found in the uploaded manuscript, section “Materials and Methods”, page 2, 3 and 4.

  1. The authors report a higher incidence of intracranial bleeding in the IUGR study population than in controls. There was also a higher incidence of platelet or plasma transfusions among IUGR neonates?

Response 2: Thank you for your comment. Indeed, we detected eight mild cases of intracranial bleeding in the IUGR study population; four neonates present with fetal subependymal hemorrhage grade I and four neonates with intraventricular hemorrhage grade I, a higher percentage when compared to control. Still, due to the small number of bleeding incidences in our study, we were not able to correlate them with the hemostatic derangement of the IUGR population. As a result, it is not clear whether those incidences are attributed to the hypocoagulable profile of IUGR neonates as noted in our study, or to the higher odds of prematurity observed in this population. As far as the platelet and plasma transfusions are concerned, the median platelet value of the IUGR population was 276,400, ranging from 231,900 to 316,100. Since our IUGR population was not thrombocytopenic and no major bleeding or need for circulation support occurred, no plasma and platelet transfusion were necessary during their stay at the NICU.

  1. The manuscript is clear enough, but sometimes it appears redundant. For example, data in Figure 1 are also presented in Table 1.

Response 3: We appreciate your feedback. Table 1 depicts the demographic, clinical and laboratory data of the study population, while Figure 1 displays the Characteristic Box & Whisker plots for the statistically significant NATEM parameters (A5, A10, A20, MCF and MCE) between IUGR and AGA neonates. However, table 2 clearly shows the data that are graphically depicted in figure 1. We hope that we can keep such redundant information, since the graphical representation allows a quick assessment of the NATEM information by the readers, while the data in table 2 can be useful for the interested readers to include them in meta-analysis studies. If it is not possible we can remove table 2 or figure 1 in a revised version.

  1. It is unclear if IUGR and AGA groups were both included in the multivariate regression analysis. How did the authors select the covariates?

Response 4: Thank you for the keen observation. We used each calculated NATEM parameter as dependent variable and the group (i.e. IUGR or AGA), Gestational age, Gender and Delivery mode as independent variables. IUGR was therefore used as an independent variable. The selected covariates represent the most common confounding factors that could affect our results. In this way we were able to accurately establish that IUGR is an independent factor causing hemostatic alterations as portrayed by NATEM variables. We have modified the statistical analysis section accordingly to clarify this.